# Effect of Exclusive Primer and Adhesive on Microtensile Bond Strength of Self-Adhesive Resin Cement to Dentin

**DOI:** 10.3390/ma13102353

**Published:** 2020-05-20

**Authors:** Bit-Na Kim, Sung-Ae Son, Jeong-Kil Park

**Affiliations:** Department of Conservative Dentistry, School of Dentistry, Pusan National University, Dental Research Institute, Yangsan 50612, Korea; 1004sjbn@hanmail.net (B.-N.K.); song-ae@hanmail.net (S.-A.S.)

**Keywords:** self-adhesive resin cement, exclusive primer, self-etching adhesive, microtensile bond strength, confocal laser scanning microscopy

## Abstract

The aim of this study was to investigate the effect of G-CEM One Primer (GCOP) and self-etching adhesive on the microtensile bond strength (µTBS) between self-adhesive resin cement G-CEM One (GCO) and dentin. Teeth were sectioned to expose the flat dentin surface and randomly assigned into five groups (n = 15) according to the dentin surface treatment: 1) no surface treatment, 2) GCOP, 3) All-Bond Universal (ABU), 4) GCOP followed by ABU (GCOP/ABU), 5) ABU followed by GCOP (ABU/GCOP). The composite resin blocks were bonded to the dentin surface using GCO. The specimens were stored in distilled water at 37 °C for 24 h, then sectioned into sticks (1 mm × 10 mm). The μTBS values were statistically analyzed using 1-way analysis of variance (ANOVA) and Tukey’s honestly significant difference (HSD) test (*α* = 0.05) and failure mode was examined under a stereomicroscope. The bonding interface of each specimen was evaluated using confocal laser scanning microscopy. The GCOP group exhibited the highest µTBS value and there were no significant differences observed between GCOP, GCOP/ABU, ABU/GCOP groups (*p* > 0.05). The use of GCOP with GCO results in the improved µTBS between GCO and dentin. In conclusion, using only GCOP with GCO for bonding of indirect restoration is extremely simple and increasing bond strength.

## 1. Introduction

Resin cement is increasingly being used in dental clinics as an adhesive material for indirect restorations. It has advantage of excellent mechanical properties, wear resistance and low solubility and marginal adaptation [1]. When bonding the indirect restoration using the conventional resin cement, the dental adhesive must be applied to the tooth surface [2]. An etch-and-rinse adhesive that demineralizes the tooth surface with removal of smear layer or self-etch adhesives dissolving smear layer can be used [3]. However, there are problems when using these adhesives; one of the disadvantages is that the process is time-consuming because of the many application stages [4]. Technique sensitivity can be induced especially when using etch-and-rinse adhesives [5].

In order to solve these problems, the self-adhesive resin cement with functional monomers was introduced. Self-adhesive resin cement can be bonded only with cement without adhesive and is recently commonly used in indirect restoration [2,6]. Self-adhesive resin cements generally include multifunctional acid methacrylate monomers containing the phosphoric acid group. The phosphoric acid group reacts with the hydroxyapatite of the tooth surface and methacrylate monomer-induced radical polymerization is initiated via a photopolymerizing or a autopolymerizing process. Therefore, a micromechanical retention occurs between the demineralized dentin and the cement. Furthermore, during polymerization, the phosphoric acid group is neutralized while it simultaneously reacts with the alkaline filler and cement shifts from hydrophilic to hydrophobic. Accordingly, the monomer bonds to the calcium ions of the hydroxyapatite resulting in a chemical retention [2,4,7,8,9].

It is important to increase the surface energy in enamel which contains high mineral content, since the formation of resin tag is essential for the micromechanical bonding [10]. Dentin, on the other hand, is hydrophilic due to its low mineral content and high proportion of organic compounds. Though infiltration of the hydrophobic resin monomer is difficult, hydrophilic and amphipathic properties of resin monomers make the bonding of dentin possible [11]. However, self-adhesive resin cement does not have consistent bond strength on the enamel [1] and dentin permeability and hydrolysis stability are not superior to the conventional resin cement using adhesive [12]. Therefore, a number of studies have reported improved bond strength through the use of adhesives before using the self-adhesive resin cement [1,12,13].

Since the longevity of an indirect restoration is directly affected by the bonding between the tooth surface and the resin cement [1], it is very important to select a product that can increase the bonding between the tooth and restoration among the diverse cements available.

Recently, G-CEM One (GCO; GC, Tokyo, Japan), a self-adhesive resin cement and GCO Primer (GCOP; GC, Tokyo, Japan), an adhesive enhancing primer were introduced. GCOP is not an adhesive, it is comprised of functional monomers and a solvent. Additionally, it includes a “touch-curing catalyst”, and can be applied not only to a tooth surface such as abutment teeth and cavities, but also to materials such as metals and resins [14,15].

According to its manufacturer, GCO achieves an improved bond strength when used together with the GCOP. However, there is insufficient research on whether using the GCO and GCOP produces strong bonding for indirect restorations such as crowns, inlays and onlays.

Therefore, the purpose of this study is to identify the effect of GCOP in terms of the μTBS between the GCO and the dentin, and to compare the strength of bonding between using self-etching adhesives in place of GCOP and using both the GCOP and the self-etching adhesives.

## 2. Materials and Methods

### 2.1. Specimen Preparation

The methods of this study were approved by the Institutional Review Board of Pusan National University Dental Hospital (IRB, PNUDH-2018–023). Healthy human molars without restorations or caries were used in this study. The extracted teeth were stored in distilled water at 4 °C for a maximum of one week. The occlusal surfaces were cut with a low-speed diamond saw (Accutom-50, Struers, Rødovre, Denmark) to expose dentin. The teeth were then polished with a polisher (Buehler Metaserv, Lake Bluff, IL, USA) and a 600-grit SiC paper for one minute to obtain a flat and uniform dentin surface covered in regular smear layer.

The surfaces of the dentin were prepared after dividing them into 5 groups. The materials used and the compositions of the materials are shown in Table 1 and each material was used in accordance with the manufacturer’s instructions. The groups were treated as follows shown in Table 2: 1) Nothing was applied to the dentin surface; 2) GCOP was applied to the dentin surface; 3) All-Bond Universal (ABU; Bisco, Schaumburg, IL, USA) was applied to the dentin surface; 4) GCOP was applied to the dentin surface followed by ABU; and 5) ABU was applied to the dentin surface followed by GCOP.

To make the resin block, 2 mm of photopolymerizing composite resin (Filtek Z-g250 A3 shade, 3 M ESPE, St. Paul, MN, USA) was incrementally filled into a cylinder-shaped plastic molding (radius 9 mm × depth 4 mm). Each layer was photopolymerized for 40 s using a 1200-mW/cm^2^ LED light curing unit (Bluephase G2, Ivoclar Vivadent, Inc., Amherst, NY, USA). Furthermore, after taking the polymerized composite resin block out of the molding, it was additionally photopolymerized on each surface for 40 s.

When bonding the composite resin block to the dentin surface, GCO mixed according to the manufacture protocol was used. A 1-N load was applied to the surface of the composite resin block to apply consistent pressure. Leftover cement was removed with a microbrush and each surface was cured with a 1200-mW/cm^2^ LED curing unit (Bluephase G2, Ivoclar Vivadent, Inc., Amherst, NY, USA) for 20 s.

### 2.2. Microtensile Bond Strength (µTBS) Testing

The specimens were stored in distilled water at 37 °C for 24 h, and then cut perpendicular to the occlusal surface of the tooth using a low-speed diamond saw (Accutom-50, Struers, Rødovre, Denmark) to form 1-mm-×-10-mm sticks. A total of 15 specimens were selected randomly from each group and each specimen was attached to the µTBS tester (Dillon Quantrol, Data Weighing Systems, Elk Grove Village, IL, USA) using a cyanoacrylate glue (Loctite, Henkel, Düsseldorf, Germany) (Figure 1). Microtensile force was applied to the specimen until fracture using a crosshead speed of 1.0 mm/min. Then, the µTBS (MPa) was measured.

### 2.3. Analysis of Failure Mode

The failure modes of fractured cross sections were analyzed using a stereomicroscope (Global A6 Series, Global surgical corporation, St. Louis, MO, USA) at 80× magnification. Failure modes were categorized as follows: cohesive failure, the failure occurred in the dentin or the composite resin; adhesive failure, the failure occurred in the interface between the dentin and the cement or between the composite resin and the cement; mixed failure, both cohesive and adhesive failures occurred.

### 2.4. Confocal Laser Scanning

Using the tooth specimen preparation methods previously described, specimens were prepared to observe the bonding interface using confocal laser scanning microscopy. To prepare the surfaces in the different groups, Rhodamine B fluorescent dye (Daejung, Seoul, Korea) was added to the ABU before applying to the dentin at a concentration of 0.01 wt%. For GCOP, fluorescein isothiocyanate (FITC) fluorescent dye (Abbkine, Wuhan, Hubei, China) was added in the concentration of 0.01 wt% and then applied to the dentin. The composite resin block was bonded using the GCO. Then, specimens were cut vertically from the occlusal surface and polished. Confocal laser scanning microscopy (LSM-700, Carl Zeiss, Oberkochen, Germany) was used to obtain images of the bonding interface in each group. The fluorescent images 100-fold magnified were analyzed using ZEN imaging software v2.6 (blue edition) (Carl Zeiss Microscopy GmbH, Jena, Germany).

### 2.5. Statistical Analysis

For statistical analyses, SPSS 23.0 (SPSS, Chicago, IL, USA) was used. The µTBS values were statistically analyzed using 1-way ANOVA and Tukey’s HSD test for post hoc pairwise comparisons was used (α = 0.05).

## 3. Results

### 3.1. μTBS

The mean and standard deviation of the µTBS in each group are shown in Table 3. Comparison of the µTBS values showed that the GCOP group had the highest values. The values were highest to lowest in the order of GCOP, GCOP/ABU, ABU/GCOP, ABU and control group. There were no statistically significant differences among the GCOP, GCOP/ABU and ABU/GCOP groups (*p* > 0.05) and there were also no statistically significant differences between ABU/GCOP and ABU groups (*p* > 0.05). ABU group showed lower µTBS values compared to the other experimental groups, but higher values compared to the control group (*p* < 0.05).

### 3.2. Analysis of Failure Mode

Table 4 shows the failure modes of specimens and Figure 2 shows the stereomicroscope image of specimens. Cohesive failures were most frequently observed in the GCOP group. In all groups other than the GCOP group, adhesive failures were the most frequently observed. Among those, adhesive failures were the most frequently observed in the control group. Furthermore, mixed failures were only observed in the GCOP and GCOP/ABU groups.

### 3.3. Confocal Laser Scanning

Figure 3 shows the confocal laser scanning microscope image of each group. The resin tag that infiltrated the dentin from the bonding interface was observed in both the GCOP group and the ABU group. Although both groups did not show a substantial difference in terms of the density or the depth of the resin tag, the GCOP was observed as a thin layer in the adhesive layer while the ABU was observed as a relatively thicker layer (Figure 3A,B). When GCOP was applied before ABU, the GCOP infiltrated the dentin and resulted in high-density GCOP in the resin tag, while ABU was limited to the adhesive layer (Figure 3C). When GCOP was applied after the ABU was applied, the ABU infiltrated the dentin and resulted in deeply infiltrated ABU in the resin tag, while GCOP was limited to the adhesive layer (Figure 3D).

## 4. Discussion

In previous studies, when using a self-adhesive resin cement, adhesive was applied to the tooth surface to increase the bond strength. However, when using GCO according to manufacturer, a GCOP that contains a “touch-curing catalyst” should be used together to increase the bond strength. Therefore, in this study, the bond strength from applying GCOP or self-etching adhesive to the tooth surface were compared. This study also compared the increase in bond strength between the tooth and the restoration when the self-etching adhesive and the GCOP are used together in different orders to that of when the self-etching adhesive or the GCOP are used alone.

Compared to the other self-etching adhesives, ABU has a relatively high pH (3.2) and is considered as an adhesive with a relatively weak hydrophilia [16,17]. Because of these characteristics, ABU does not cause a problem of incompatibility with photopolymerizing or autopolymerizing resin cements [18]. For this reason, ABU was used to compare with GCOP in this study. The ABU group showed higher µTBS values compared to the control group in this study (*p* < 0.05). These results are consistent with the results of previous studies which reported that the treated group with self-etching adhesive before using self-adhesive resin cement showed a higher µTBS than the group without self-etching adhesive [1,12,13].

As both self-adhesive resin cement and self-etching adhesive contain functional monomer, they can demineralize and infiltrate tooth substrate making micromechanical retention [2,13,19]. The functional monomer 10-methacryloyloxydecyl dihydrogen phosphate (MDP) forms a strong bond with the hydroxyapatite through the MDP–Ca salt, and has a lower solubility compared to other monomers [20]. Self-adhesive cements, on the other hand, have low flowability due to their viscosity and only work on the dental hard tissue surface, resulting in limited infiltration to the dentin [21]. In addition, some studies have reported that no hybrid layer or resin tag was observed when using only self-adhesive resin cement. Therefore, the bonding is weak when only the self-adhesive cement is used [22,23]. However, when self-etching adhesives are used, resin monomers can infiltrate sufficiently into the dentin improving bond strength. In other words, before applying a self-adhesive resin cement, the use of adhesives with 10-MDP such as ABU results in higher bond strength [24,25] and also improves resistance to degradation and longevity of bonding [19].

In addition, in order to improve bonding in the dentin, it is important to protect the collagen exposed in the hybrid layer. The interaction between the monomer and the collagen has an impact on the collagen [26]. MDP interacts with the hydroxyapatite to form MDP–Ca. Moreover, the hydrophobic part of the MDP has a hydrophobic interaction with the collagen to protect the collagen [27]. The composition of ABU includes 4-methacryloyloxyethyl trimellitic anhydride (4-META), 10-MDP and hydroxyethyl methacrylate (HEMA). When MDP and HEMA are combined, HEMA reacts with the hydrophilic part of the MDP to form the MDP–HEMA complex. Thus, the possibility of interaction between the hydrophobic part of the MDP in the center of the complex and the collagen is decreased. In other words, MDP–HEMA weakens the hydrophobic interaction between MDP and collagen, resulting in the lack of collagen protection [27,28,29]. Therefore, the bonding between the dentin and the cement is relatively weakened when ABU is used compared to when GCOP, which has 10-MDP, but not HEMA, is used. For this reason, in the results of this study, the GCOP group showed the highest µTBS values, and showed statistically significant differences from both the control group and the ABU group (*p* < 0.05).

GCOP additionally includes a “touch-curing catalyst” in addition to 10-MDP, which is a functional monomer. It is difficult for a clinically sufficient light source to reach the bonding interface between the dentin and the restoration, Therefore, “touch curing” plays an important role in the improvement of the bond strength of the resin cement [30]. In other words, because there can be a chemical polymerization when the primer and the cement come in contact through “touch curing,” the bond strength between the cement and the dentin is increased [31]. Therefore, in terms of the failure mode of the specimens, the lowest frequency of adhesive failure was observed in the GCOP group compared to the other groups (Table 4). This corresponds with the results of GCOP showing the highest µTBS values (Table 3).

In this study, both GCOP and ABU were used by applying the two materials in different orders. Both the GCOP/ABU group and the ABU/GCOP group showed higher µTBS values compared to the control group (*p* < 0.05). The confocal laser scanning microscope images of this study showed results similar to those of a previous study [30], which compared the strength of photopolymerization-dependent bonding whether the curing was done on each of self-etching adhesives or resin cement. When GCOP was applied, air dried and ABU was photopolymerized, a layer comprising of a combination of the two materials would were formed (Figure 3C) and the bond strength to dentin would have improved due to the 10-MDP. Furthermore, when applying ABU, photopolymerizing and then applying GCOP, the infiltration of the GCOP to the dentin would were impossible due to the polymerized ABU, thus resulting in separate layers (Figure 3D), with the effect of the “touch-curing catalyst” showing on the upper layer of GCOP.

Although among the GCOP, GCOP/ABU and ABU/GCOP groups, the µTBS values increased in the order of GCOP, GCOP/ABU and ABU/GCOP, there were no statistically significant differences (*p* > 0.05). This signifies that the increasing of the bonding to dentin with GCOP containing 10-MDP and to cement with GCOP containing the “touch-curing catalyst” are similar. It also shows that the bond strength of either bonding is as strong as both of them.

Furthermore, the GCOP/ABU group showed higher µTBS values compared to those of the ABU group (*p* < 0.05). This suggested that the 10-MDP of the additional GCOP in the GCOP/ABU group can improve the bonding to dentin although MDP–HEMA complex is formed in ABU.

However, statistically significant differences were not observed between the ABU/GCOP and the ABU group (*p* > 0.05). Thus, the weakening of the interaction between the dentin and the cement due to the formation of the MDP–HEMA complex when ABU is first applied to the tooth surface is avoidable if the “touch-curing catalyst” in GCOP increasing bond strength is additionally applied.

The results of this study showed that using GCOP when the indirect restorations are bonded using the GCO can improve the bond strength between the dentin and the restoration. Using self-etching adhesives instead of GCOP results improved bond strength compared to when the cement is used alone, but the effect of improvement of bonding is small compared to using the GCOP. Furthermore, when GCOP was used, there were no differences in the improvement of bonding regardless of whether additional self-etching adhesives were used.

This in vitro research is limited in that it could not represent variations in the real oral environments. Therefore, long-term research that reflects variations in clinical environments should be additionally performed.

## 5. Conclusions

Despite the limitations of this study, the conclusion is that using GCOP when bonding an indirect restoration with GCO improves bond strength, while using additional self-etching adhesives does not affect the bond strength. Therefore, when bonding indirect restorations using the GCO, using only the GCOP can be an effective method that can produce improved bonding to the dentin with fewer technical steps and shorter time necessary.

## Figures and Tables

**Figure 1 materials-13-02353-f001:**
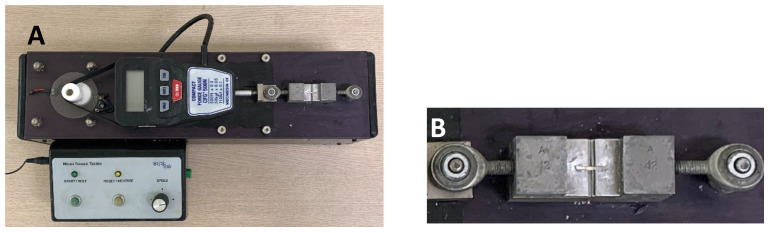
(**A**) µTBS tester; (**B**) specimen is mounted on the specimen holder.

**Figure 2 materials-13-02353-f002:**
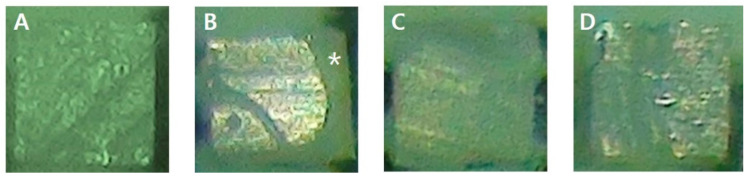
Stereomicroscope images of debonded specimens. (**A**) adhesive failure (dentin side); (**B**) mixed failure, the resin in the dentin side could be seen (*: asterisk); (**C**) cohesive failure in dentin; (**D**) cohesive failure in resin.

**Figure 3 materials-13-02353-f003:**
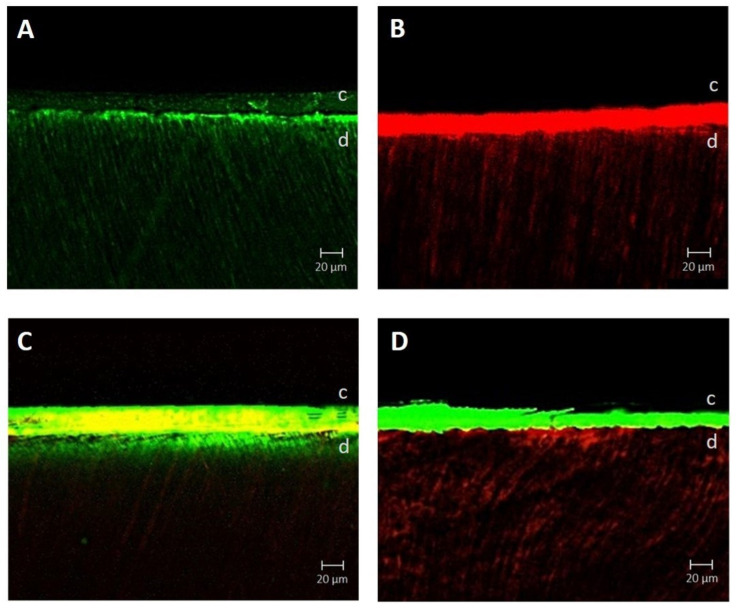
Confocal laser scanning microscopy images of groups with different surface treatment. (**A**) of G-CEM One Primer (GCOP); (**B**) All-Bond Universal (ABU); (**C**) GCOP followed by ABU; (**D**) ABU followed by GCOP. The green fluorescent signal is GCOP dyed with fluorescein isothiocyanate (FITC) and the red fluorescent signal is ABU dyed with Rhodamine B. In addition, the yellow fluorescent signal is a layer comprising of a combination of the GCOP and ABU. * c = cement, d = dentin.

**Table 1 materials-13-02353-t001:** Manufacturer and composition of materials used.

Material	Manufacturer	Composition
G-CEM One Primer	GC Corp., Tokyo, Japan	ethanol, 10-MDP, 10-methacryloyloxydecyl dihydrogen thiophosphate, 4-META, 2-hydroxy-1,3 dimethoxypropane, vanadyl acetylacetonate, 2,6-di-tert-butyl-*p*-cresol
All-Bond Universal	Bisco, Schaumburg, IL, USA	10-MDP, 2-HEMA, Bis-GMA, ethanol, water, photoinitiator
G-CEM One	GC Corp., Tokyo, Japan	Paste A: fluoroaluminosilicate glass, UDMA, dimethacrylate, initiator, stabilizer, pigment, silicon dioxide, MDP Paste B: SiO_2_, trimethoxysilane, UDMA, 2-hydroxy-1,3-dimethacryloxypropane, MDP, 6-tert-butyl-2,4-xylenol, 2,6-di-tert-butyl-*p*-cresol, EDTA disodium salt dehydrate, vanadyl acetylacetonate, TPO, ascorbic acid, camphorquinone, MgO

MDP: 10-methacryloyloxydecyl dihydrogen phosphate, 4-META: 4-methacryloyloxyethyl trimellitic anhydride, HEMA: 2-hydroxyethyl methacrylate, Bis-GMA: 2,2-Bis[4-(2-hydroxy-3-methacryloxypropoxy)phenyl]propane, UDMA: Urethane dimethacrylate, EDTA: ethylenediaminetetraacetic acid, TPO: thermoplastic polyolefin.

**Table 2 materials-13-02353-t002:** Surface treatment and application method of materials used in the groups.

Groups	Surface treatment	Application Procedure
Control	No surface treatment	
GCOP	G-CEM One Primer	Apply on dentin surface and rubbing with a microbrush for 10 s then air dry for 5 s
ABU	All-Bond Universal	Apply on dentin surface and air dry to remove excess solvent then light curing for 10 s
GCOP/ABU	G-CEM One Primer followed by All-Bond Universal	Apply G-CEM One Primer followed by All-Bond Universal in the same way above.
ABU/GCOP	All-Bond Universal followed by G-CEM One Primer	Apply All-Bond Universal followed by G-CEM One Primer in the same way above.

**Table 3 materials-13-02353-t003:** Microtensile bond strength values (MPa) means and standard deviation.

Groups	µTBS
Control	9.2 (2.2)^a^
GCOP	20.1 (5.5)^b^
ABU	13.9 (1.7)^c^
GCOP/ABU	18.1 (2.3)^b^
ABU/GCOP	16.9 (4.4)^b,c^

Different superscript letters (^a,b,c^) indicate statistically differences within the same column (*p* < 0.05).

**Table 4 materials-13-02353-t004:** Failure mode distribution in each group (%). A = adhesive, M = mixed, DC = dentin cohesive, RC = resin cohesive.

Groups	A	M	RC	DC
Control	11 (73.3)	0	1 (6.7)	3 (20)
GCOP	6 (40)	1 (6.7)	2 (13.3)	6 (40)
ABU	9 (60)	0	2 (13.3)	4 (26.7)
GCOP/ABU	9 (60)	1 (6.7)	0	5 (33.3)
ABU/GCOP	10 (66.7)	0	0	5 (33.3)

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
