# Peer review of "Effect of Exclusive Primer and Adhesive on Microtensile Bond Strength of Self-Adhesive Resin Cement to Dentin"

_materials, 2020, doi:10.3390/ma13102353_

Round 1

Reviewer 1 Report

Overall the paper appears satisfactory regarding experimental design, protocol, results and conclusions. A couple of items should be addressed. First, the reason that GCOP addition to ABU (ABU/GCOP) improves bond strength over ABU alone is not clear to this reviewer. Since ABU has already penetrated the dentin prior to GCOP application, why would GCOP improve adhesion? Secondly, a bit more detail should be provided for the confocal images – the reader needs to know what red and green fluorescence is showing, which side is dentin (top or bottom), and more magnified images are needed to truly observe dentin penetration and how ABU and GCOP appear when used together in either combination.

Author Response

Reviewer 1

Overall the paper appears satisfactory regarding experimental design, protocol, results and conclusions. A couple of items should be addressed.

First, the reason that GCOP addition to ABU (ABU/GCOP) improves bond strength over ABU alone is not clear to this reviewer. Since ABU has already penetrated the dentin prior to GCOP application, why would GCOP improve adhesion?

ànswer

238 However, statistically significant differences were not observed between the ABU/GCOP and the ABU group (p > 0.05). Thus, the weakening of the interaction between the dentin and the cement due to the formation of the MDP-HEMA complex when ABU is first applied to the tooth surface is avoidable if the touch curing catalyst in GCOP increasing bond strength is additionally applied.

ABU has already penetrated the dentin prior to GCOP application, GCOP improve adhesion because of touch curing catalyst. I modified the sentence (252 row)

Secondly, a bit more detail should be provided for the confocal images – the reader needs to know what red and green fluorescence is showing, which side is dentin (top or bottom), and more magnified images are needed to truly observe dentin penetration and how ABU and GCOP appear when used together in either combination.

ànswer

I made the figure bigger and dentin and cement side are marked.

Reviewer 2 Report

This is an excellent study that only requires Extensive editing of English language and style.  Aside from grammatical errors, there are quite a few sentences that do not really make sense in English.  Also, I am used to reading the Methods section before the results but perhaps this is particular to this journal (Materials).

Author Response

Reviewer 2

This is an excellent study that only requires Extensive editing of English language and style.  Aside from grammatical errors, there are quite a few sentences that do not really make sense in English.  Also, I am used to reading the Methods section before the results but perhaps this is particular to this journal (Materials).

ànswer

This manuscript was proofread and edited by professional English editor.

I attached the CERTIFICATE OF TRANSLATION

Reviewer 3 Report

The manuscript  Effect of Exclusive Primer and Adhesive on Microtensile Bond Strength of Self-Adhesive Resin Cement to Dentin must be seriously improved by the authors before publication.

The purpose of this study is to identify the effect of GCOP in terms of the bond strength between the GCO and the dentin, and to compare the strength of bonding between using  self-etching adhesives in place of GCOP and using both the GCOP and the self-etching adhesives. My comments and recommendation are mentioned below:

  1. The introduction is too short. Some details about the aspects correlated with the study (adhesion in dentistry, bond strenght, methods used for testing) are missing in introduction. The aspects about adhesion is missing in the introduction.
  2. The pharagraph dedicated to the GCO product include many sentences who is not supported by references. Could be nice to include some references or reformulate.
  3. Part 4 (Materials and methods) must be moved as part 2 (between 1. Introduction and 3. Results).
  4. Authors could present the stereomicroscopy results (as images), for all samples.
  5. I think that could be useful if the authors will present a images with the experimental samples positioned for μTBS testing.
  6. The results regarding Percentage (%) of failure mode in each group is not very well presented. I suggest to be more clearly and specify exactly the values, and after that present the percentage as complementary interpretation of these results. Also, the authors must present some images (stereomicroscopy or SEM) for each type of failure (cohesive, adhesive, and especially for the mix).
  7. The main limitation of the study is the absence of surface characterization before bonding. The authors mention that the surfaces of the dentin were prepared differently (control and 4 different surface treatments) , but they not performed any surface characterization regarding the surface roughness. The surface quality influence strongly the bonding and must be presented minimal data about these values, in order to be correlated with μTBS results and failure modes.
  8. Even is not mandatory for this study because they use stereomicroscopy, a SEM analysis of the different failures of the experimental samples could be more interesting for the readers.
  9. The level of references must be improved. I suggest some papers from the journal dedicated to adhesion science and technology, or dental adhesives.

Author Response

Reviewer 3

The purpose of this study is to identify the effect of GCOP in terms of the bond strength between the GCO and the dentin, and to compare the strength of bonding between using  self-etching adhesives in place of GCOP and using both the GCOP and the self-etching adhesives. My comments and recommendation are mentioned below:

1.The introduction is too short. Some details about the aspects correlated with the study (adhesion in dentistry, bond strenght, methods used for testing) are missing in introduction. The aspects about adhesion is missing in the introduction.

ànswer

I revised the introduction and added sentences (enamel & dentin bonding, adhesive system) .

2. The pharagraph dedicated to the GCO product include many sentences who is not supported by references. Could be nice to include some references or reformulate.

ànswer

I included references.

3.Part 4 (Materials and methods) must be moved as part 2 (between 1. Introduction and 3. Results).

ànswer

Part 4 moved as part 2.

4. Authors could present the stereomicroscopy results (as images), for all samples.

ànswer

I included the image of stereomicroscopy (figure 2)

5. I think that could be useful if the authors will present a images with the experimental samples positioned for μTBS testing.

ànswer

I included a picture (Figure 1)

6. The results regarding Percentage (%) of failure mode in each group is not very well presented. I suggest to be more clearly and specify exactly the values, and after that present the percentage as complementary interpretation of these results.

ànswer

I suggested values (%) to be more clear.

Also, the authors must present some images (stereomicroscopy or SEM) for each type of failure (cohesive, adhesive, and especially for the mix).

 ànswer

I included the image of stereomicroscopy (figure 2).

7. The main limitation of the study is the absence of surface characterization before bonding. The authors mention that the surfaces of the dentin were prepared differently (control and 4 different surface treatments) , but they not performed any surface characterization regarding the surface roughness. The surface quality influence strongly the bonding and must be presented minimal data about these values, in order to be correlated with μTBS results and failure modes.

ànswer

Before adhesion test, all specimens were polished with 600 grit SiC paper. And this sentence was included in the materials and methods

8. Even is not mandatory for this study because they use stereomicroscopy, a SEM analysis of the different failures of the experimental samples could be more interesting for the readers.

ànswer

I included the image of stereomicroscopy instead of SEM.

9. The level of references must be improved. I suggest some papers from the journal dedicated to adhesion science and technology, or dental adhesives.

ànswer

Some references were changed. The level of references was improved.

Round 2

Reviewer 3 Report

The authors perform more of suggested improvements. But they still don't answer to the main problem.

The main limitation of the study is the absence of surface characterization before bonding. The authors mention that the surfaces of the dentin were prepared differently (control and 4 different surface treatments) , but they not performed any surface characterization regarding the surface roughness. The surface quality influence strongly the bonding and must be presented minimal data about these values, in order to be correlated with μTBS results and failure modes.

Their answer "Before adhesion test, all specimens were polished with 600 grit SiC paper. And this sentence was included in the materials and methods" didn't give any information about the surface properties.

Round 3

Reviewer 3 Report

According the authors reply, looks like they don't understand or don't want to understand the problem. I don't agree the author comment "dentin surface quality influencing bonding is considered negligible ". Also, they mention "Since all specimen in the all groups were tested under the same surface conditions" but didn't bring any experiments for that or shown any results. So, I still considering that the manuscript must be improved before publication.

The main limitation of the study is the absence of surface characterization before bonding. The authors mention that the surfaces of the dentin were prepared differently (control and 4 different surface treatments) , but they not performed any surface characterization regarding the surface roughness. The surface quality influence strongly the bonding and must be presented minimal data about these values, in order to be correlated with μTBS results and failure modes.
Their answer "Before adhesion test, all specimens were polished with 600 grit SiC paper. And this sentence was included in the materials and methods" didn't give any information about the surface properties.